# Reinforcement Learning of Theorem Proving

**Cezary Kaliszyk**[*]
University of Innsbruck

**Josef Urban**[*]
Czech Technical University in Prague

**Henryk Michalewski**
University of Warsaw,
Institute of Mathematics of the
Polish Academy of Sciences,
deepsense.ai

**Mirek Olšák**
Charles University

## Abstract

We introduce a theorem proving algorithm that uses practically no domain heuristics for guiding its connection-style proof search. Instead, it runs many Monte-Carlo simulations guided by reinforcement learning from previous proof attempts. We produce several versions of the prover, parameterized by different learning and guiding algorithms. The strongest version of the system is trained on a large corpus of mathematical problems and evaluated on previously unseen problems. The trained system solves within the same number of inferences over 40% more problems than a baseline prover, which is an unusually high improvement in this hard AI domain. To our knowledge this is the first time reinforcement learning has been convincingly applied to solving general mathematical problems on a large scale.

## 1 Introduction

Automated theorem proving (ATP) [38] can in principle be used to attack any formally stated mathematical problem. For this, state-of-the-art ATP systems rely on fast implementations of complete proof calculi such as resolution [37], superposition [4], SMT [5] and (connection) tableau [15] that have been over several decades improved by many search heuristics. This is already useful for automatically discharging smaller proof obligations in large interactive theorem proving (ITP) verification projects [7]. In practice, today's best ATP system are however still far weaker than trained mathematicians in most research domains. Machine learning from many proofs could be used to improve on this.

Following this idea, large formal proof corpora have been recently translated to ATP formalisms [45, 32, 19], and machine learning over them has started to be used to train guidance of ATP systems [47, 28, 2]. First, to select a small number of relevant facts for proving new conjectures over large formal libraries [1, 6, 11], and more recently also to guide the internal search of the ATP systems. In sophisticated saturation-style provers this has been done by feedback loops for strategy invention [46, 17, 39] and by using supervised learning [16, 30] to select the next given clause [31]. In the simpler connection tableau systems such as leanCoP [34], supervised learning has been used to choose the next tableau extension step [48, 20] and first experiments with Monte-Carlo guided proof search [10] have been done. Despite a limited ability to prioritize the proof search, the guided search in the latter connection systems is still organized by *iterative deepening*. This ensures completeness, which has been for long time a *sine qua non* for building proof calculi.

In this work, we remove this requirement, since it basically means that all shorter proof candidates have to be tried before a longer proof is found. The result is a bare connection-style theorem prover

---

[*]These authors contributed equally to this work.

that does not use any human-designed proof-search restrictions, heuristics and targeted (decision) procedures. This is in stark contrast to recent mainstream ATP research, which has to a large extent focused on adding more and more sophisticated human-designed procedures in domains such as SMT solving.

Based on the bare prover, we build a sequence of systems, adding Monte-Carlo tree search [25], and reinforcement learning [44] of policy and value guidance. We show that while the performance of the system (called rlCoP) is initially much worse than that of standard leanCoP, after ten iterations of proving and learning it solves significantly more previously unseen problems than leanCoP when using the same total number of inference steps.

The rest of the paper is organized as follows. Section 2 explains the basic connection tableau setting and introduces the bare prover. Section 3 describes integration of the learning-based guiding mechanisms, i.e. Monte-Carlo search, the policy and value guidance. Section 4 evaluates the system on a large corpus of problems extracted from the Mizar Mathematical Library [13].

## 2 The Game of Connection Based Theorem Proving

We assume basic first-order logic and theorem proving terminology [38]. We start with the connection tableau architecture as implemented by the leanCoP [34] system. leanCoP is a compact theorem prover whose core procedure can be written in seven lines in Prolog. Its input is a (mathematical) problem consisting of *axioms* and *conjecture* formally stated in first-order logic (FOL). The calculus searches for *refutational proofs*, i.e. proofs showing that the axioms together with the negated conjecture are *unsatisfiable*.[2] The FOL formulas are first translated to the *clause normal form* (CNF), producing a set of first-order *clauses* consisting of *literals* (atoms or their negations). An example set of clauses is shown in Figure 1. The figure also shows a *closed connection tableau*, i.e., a finished proof tree where every branch contains *complementary literals* (literals with opposite polarity). Since all branches contain a pair of contradictory literals, this shows that the set of clauses is unsatisfiable.

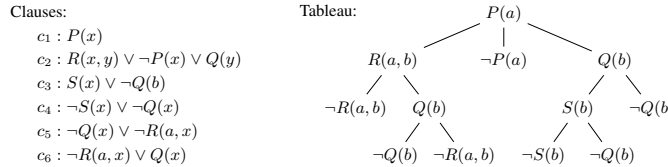

Figure 1: Closed connection tableau for a set of clauses (adapted from Letz et al. [29]).

The proof search starts with a *start clause* as a *goal* and proceeds by building a connection tableau by repeatedly applying *extension steps* and *reduction steps* to it. The extension step connects (*unifies*) the *current goal* (a selected tip of a tableau branch) with a complementary literal of a new clause. This extends the *current branch*, possibly splitting it into several branches if there are more literals in the new clause, and possibly *instantiating* some variables in the tableau. The reduction step connects the current goal to a complementary literal of the *active path*, thus *closing* the current branch. The proof is finished when all branches are closed. The extension and reduction steps are nondeterministic, requiring backtracking in the standard connection calculus. *Iterative deepening* is typically used to ensure completeness, i.e. making sure that the proof search finds a proof if there is any. *Incomplete strategies* that restrict backtracking can be used in leanCoP, sometimes improving its performance on benchmarks [33].

### 2.1 The Bare Prover

Our bare prover is based on a previous reimplementation [23] of leanCoP in OCaml (*mlCoP*). Unlike the Prolog version, mlCoP uses an explicit stack for storing the full proof state. This allows

us to use the full proof state for machine learning guidance. We first modify mlCoP by removing iterative deepening, i.e., the traversal strategy that makes sure that shorter (shallower) tableaux are tested before deeper ones. Instead, the bare prover randomly chooses extension and reduction steps operating on the current goal, possibly going into arbitrary depth. This makes our bare prover trivially incomplete. A simple example to demonstrate that is the unsatisfiable set of clauses $\{P(0), \neg P(x) \vee P(s(x)), \neg P(0)\}$ . If the prover starts with the goal $P(0)$, the third clause can be used to immediately close the tableau. Without any depth bounds, the bare prover may however also always extend with the second clause, generating an infinite branch $P(0), P(s(0)), P(s(s(0))), ....$ Rather than designing such completeness bounds and corresponding exhaustive strategies, we will use Monte-Carlo search and reinforcement learning to gradually teach the prover to avoid such bad branches and focus on the promising ones.

Next, we add playouts and search node visit counts. A *playout* of length $d$ is simply a sequence of $d$ consecutive extension/reduction steps (*inferences*) from a given *proof state* (a tableau with a selected goal). Inferences thus correspond to *actions* and are similar to moves in games. We represent inferences as integers that encode the selected clause together with the literal that connected to the goal. Instead of running one potentially infinite playout, the bare prover can be instructed to play $n$ playouts of length $d$. Each playout updates the counts for the *search nodes* that it visits. Search nodes are encoded as sequences of inferences starting at the empty tableau. A playout can also run without length restrictions until it visits a previously unexplored search node. This is the current default. Each playout in this version always starts with empty tableau, i.e., it starts randomly from scratch.

The next modification to this simple setup are *bigsteps* done after $b$ playouts. They correspond to moves that are chosen in games after many playouts. Similarly, instead of starting all playouts always from scratch (empty tableau) as above, we choose after the first $b$ playouts a particular single inference (bigstep), resulting in a new *bigstep tableau*. The next $b$ playouts will start with this tableau, followed by another bigstep, etc.

This finishes the description of the bare prover. Without any guidance and heuristics for choosing the playout inferences and bigsteps, this prover is typically much weaker than standard mlCoP with iterative deepening, see Section 4. The bare prover will just iterate between randomly doing $b$ new playouts, and randomly making a bigstep.

## 3 Guidance

The rlCoP extends the bare prover with (i) Monte-Carlo tree search balancing exploration and exploitation using the UCT formula [25], (ii) learning-based mechanisms for estimating the prior probability of inferences to lead to a proof (*policy*), and (iii) learning-based mechanisms for assigning heuristic *value* to the proof states (tableaux).

### 3.1 Monte-Carlo Guidance

To implement Monte-Carlo tree search, we maintain at each search node $i$ the number of its visits $n_i$, the total reward $w_i$, and its prior probability $p_i$. This is the transition probability of the action (inference) that leads from $i$'s parent node to $i$. If no policy learning is used, the prior probabilities are all equal to one. The total reward for a node is computed as a sum of the rewards of all nodes below that node. In the basic setting, the reward for a leaf node is $1$ if the sequence of inferences results in a closed tableau, i.e., a proof of the conjecture. Otherwise it is $0$.

Instead of this basic setting, we will by default use a simple evaluation heuristic, that will later be replaced by learned value. The heuristic is based on the number of open (non-closed) goals (tips of the tableau) $G_o$. The exact value is computed as $0.95^{G_o}$, i.e., the leaf value exponentially drops with the number of open goals in the tableau. The motivation is similar as, e.g., preferring smaller clauses (closer to the empty clause) in saturation-style theorem provers. If nothing else is known and the open goals are assumed to be independent, the chances of closing the tableau within a given inference limit drop exponentially with each added open goal. The exact value of $0.95$ has been determined experimentally using a small grid search.

We use the standard UCT formula [25] to select the next inferences (actions) in the playouts:

$$\frac{w_i}{n_i} + c \cdot p_i \cdot \sqrt{\frac{\ln N}{n_i}}$$

where $N$ stands for the total number of visits of the parent node. We have also experimented with PUCT as in AlphaZero [43], however the results are practically the same. The value of $c$ has been experimentally set to 2 when learned policy and value are used.

## 3.2 Policy Learning and Guidance

From many proof runs we learn prior probabilities of actions (inferences) in particular proof states corresponding to the search nodes in which the actions were taken. We characterize the proof states for policy learning by extracting features (see Section 3.4) from the current goal, the active path, and the whole tableau. Similarly, we extract features from the clause and its literal that were used to perform the inference. Both are extracted as sparse vectors and concatenated into pairs $(f_{state}, f_{action})$. For each search node, we extract from its UCT data the frequency of each action $a$, and normalize it by dividing with the average action frequency at that node. This yields a relative proportion $r_a \in (0, \infty)$. Each concatenated pair of feature vectors $(f_s, f_a)$ is then associated with $r_a$, which constitutes the training data for policy learning implemented as regression on the logarithms. During the proof search, the prior probabilities $p_i$ of the available actions $a_i$ in a state $s$ are computed as a softmax of their predictions. We use $\tau = 2.5$ by default as the softmax temperature. This value has been optimized by a small grid search.

The policy learning data can be extracted from all search nodes or only from some of them. By default, we only extract the training examples from the bigstep nodes. This makes the amount of training data manageable for our experiments and also focuses on important examples.

## 3.3 Value Learning and Guidance

Bigstep nodes are used also for learning of the proof state evaluation (value). For value learning we characterize the proof states of the nodes by extracting features from all goals, the active path, and the whole tableau. If a proof was found, each bigstep node $b$ is assigned value $v_b = 1$. If the proof search was unsuccessful, each bigstep is assigned value $v_b = 0$. By default we also apply a small discount factor to the positive bigstep values, based on their distance $d_{proof}(b)$ to the closed tableau, measured by the number of inferences. This is computed as $0.99^{d_{proof}(b)}$. The exact value of $0.99$ has again been determined experimentally using a small grid search.

For each bigstep node $b$ the sparse vector of its proof state features $f_b$ is associated with the value $v_b$. This constitutes the training data for value learning which is implemented as regression on the logits. The trained predictor is then used during the proof search to estimate the logit of the proof state value.

## 3.4 Features

We have briefly experimented with using deep neural networks to learn the policy and value predictors directly from the theorem proving data. Current deep neural architectures however do not seem to perform on such data significantly better than non-neural classifiers such as XGBoost with manually engineered features [16, 36, 2]. We use the latter approach, which is also significantly faster [30].

Features are collected from the first-order terms, clauses, goals and tableaux. Most of them are based on (normalized) term walks of length up to 3, as used in the ENIGMA system [16]. These features are in turn based on the syntactic and semantic features developed in [24, 47, 19, 18, 21]. We uniquely identify each symbol by a 64-bit integer. To combine a sequence of integers originating from symbols in a term walk into a single integer, the components are multiplied by fixed large primes and added. The resulting integers are then reduced to a smaller feature space by taking modulo by a large prime ($2^{18} - 5$). The value of each feature is the sum of its occurrences in the given expression.

In addition to the term walks we also use several common abstract features, especially for more complicated data such as tableaux and paths. Such features have been previously used for learning

strategy selection [27, 40]. These are: number of goals, total symbol size of all goals, maximum goal size, maximum goal depth, length of the active path, number of current variable instantiations, and the two most common symbols and their frequencies. The exact features used have been optimized based on several experiments and analysis of the reinforcement learning data.

### 3.5 Learners and Their Integration

For both policy and value we have experimented with several fast linear learners such as LIBLIN-EAR [9] and the XGBoost [8] gradient boosting toolkit (used with the linear regression objective). The latter performs significantly better and has been chosen for conducting the final evaluation. The XGBoost parameters have been optimized on a smaller dataset using randomized cross-validated search[3] taking speed of training and evaluation into account. The final values that we use both for policy and value learning are as follows: maximum number of iterations $= 400$, maximum tree depth $= 9$, ETA (learning rate) $= 0.3$, early stopping rounds $= 200$, lambda (weight regularization) $= 1.5$. Table 1 compares the performance of XGBoost and LIBLINEAR[4] on value data extracted from 2003 proof attempts.

Table 1: Machine learning performance of XGBoost and LIBLINEAR on the value data extracted from 2003 problems. The errors are errors on the logits.

| Predictor | Train Time | RMSE (Train) | RMSE (Test) |
|---|---|---|---|
| XGBoost | 19 min | 0.99 | 2.89 |
| LIBLINEAR | 37 min | 0.83 | 16.31 |

For real-time guidance during the proof search we have integrated LIBLINEAR and XGBoost into rlCoP using the OCaml foreign interface, which allows for a reasonably low prediction overhead. Another part of the guidance overhead is feature computation and transformation of the computed feature vectors into the form accepted by the learned predictors. Table 2 shows that the resulting slowdown is in low linear factors.

Table 2: Inference speed comparison of mlCoP and rlCoP. IPS stand for inferences per second. The data are averaged over 2003 problems.

| System | mlCoP | rlCoP without policy/value (UCT only) | rlCoP with XGBoost policy/value |
|---|---|---|---|
| Average IPS | 64335.5 | 64772.4 | 16205.7 |

## 4 Experimental Results

The evaluation is done on two datasets of first-order problems exported from the Mizar Mathematical Library [13] by the MPTP system [45]. The larger *Miz40* dataset [5] consists of 32524 problems that have been proved by several state-of-the-art ATPs used with many strategies and high time limits in the experiments described in [21]. See Section 4.6 for a discussion of the performance of these systems on *Miz40*. We have also created a smaller *M2k* dataset by taking 2003 Miz40 problems that come from related Mizar articles. Most experiments and tuning were done on the smaller dataset. All problems are run on the same hardware[6] and with the same memory limits. When using UCT, we always run 2000 playouts (each until a new node is found) per bigstep.

### 4.1 Performance without Learning

First, we use the M2k dataset to compare the performance of the baseline mlCoP with the bare prover and with the non-learning rlCoP using only UCT with the simple goal-counting proof state evaluation heuristic. The results of runs with a limit of 200000 inferences are shown in Table 3.

Raising the inference limit helps only a little: mlCoP solves $1003$ problems with a limit of $2*10^6$ inferences, and $1034$ problems with a limit of $4*10^6$ inferences. The performance of the bare prover is low as expected - only about half of the performance of mlCoP. rlCoP using UCT with no policy and only the simple proof state evaluation heuristic is also weaker than mlCoP, however already significantly better than the bare prover.

Table 3: Performance on the M2k dataset of mlCoP, the bare prover and non-learning rlCoP with UCT and simple goal-counting proof state evaluation. (200000 inference limit).

| System | mlCoP | bare prover | rlCoP without policy/value (UCT only) |
|---|---|---|---|
| Problems proved | 876 | 434 | 770 |

## 4.2 Reinforcement Learning of Policy Only

Next we evaluate on the M2k dataset rlCoP with UCT using only policy learning, i.e., the value is still estimated heuristically. We run 20 iterations, each with 200000 inference limit. After each iteration we use the policy training data (Section 3.2) from all previous iterations to train a new XGBoost predictor. This is then used for estimating the prior action probabilities in the next run. The 0th run uses no policy. This means that it is the same as in Section 4.1, solving 770 problems. Table 4 shows the problems solved by iterations 1 to 20. Already the first iteration significantly improves over mlCoP run with 200000 inference limit. Starting with the fourth iteration, rlCoP is better than mlCoP run with the much higher $4*10^6$ inference limit.

Table 4: 20 policy-guided iterations of rlCoP on the M2k dataset.

| Iteration | 1 | 2 | 3 | 4 | 5 | 6 | 7 | 8 | 9 | 10 |
|---|---|---|---|---|---|---|---|---|---|---|
| Proved | 974 | 1008 | 1028 | 1053 | 1066 | 1054 | 1058 | 1059 | 1075 | 1070 |
| Iteration | 11 | 12 | 13 | 14 | 15 | 16 | 17 | 18 | 19 | 20 |
| Proved | 1074 | 1079 | 1077 | 1080 | 1075 | 1075 | **1087** | 1071 | 1076 | 1075 |

## 4.3 Reinforcement Learning of Value Only

Similarly, we evaluate on the M2k dataset 20 iterations of rlCoP with UCT and value learning, but with no learned policy (i.e., all prior inference probabilities are the same). Each iteration again uses a limit of 200000 inferences. After each iteration a new XGBoost predictor is trained on the value data (Section 3.3) from all previous iterations, and is used to evaluate the proof states in the next iteration. The 0th run again uses neither policy nor value, solving 770 problems. Table 5 shows the problems solved by iterations 1 to 20. The performance nearly reaches mlCoP, however it is far below rlCoP using policy learning.

Table 5: 20 value-guided iterations of rlCoP on the M2k dataset.

| Iteration | 1 | 2 | 3 | 4 | 5 | 6 | 7 | 8 | 9 | 10 |
|---|---|---|---|---|---|---|---|---|---|---|
| Proved | 809 | 818 | 821 | 821 | 818 | 824 | **856** | 831 | 842 | 826 |
| Iteration | 11 | 12 | 13 | 14 | 15 | 16 | 17 | 18 | 19 | 20 |
| Proved | 832 | 830 | 825 | 832 | 828 | 820 | 825 | 825 | 831 | 815 |

## 4.4 Reinforcement Learning of Policy and Value

Finally, we run on the M2k dataset 20 iterations of full rlCoP with UCT and both policy and value learning. The inference limits, the 0th run and the policy and value learning are as above. Table 6 shows the problems solved by iterations 1 to 20. The 20th iteration proves 1235 problems, which is $19.4\%$ more than mlCoP with $4*10^6$ inferences, $13.6\%$ more than the best iteration of rlCoP with policy only, and $44.3\%$ more than the best iteration of rlCoP with value only. The first iteration improves over mlCoP with 200000 inferences by $18.4\%$ and the second iteration already outperforms the best policy-only result.

We also evaluate the effect of joint reinforcement learning of policy and value. Replacing the final policy with the best one from the policy-only runs decreases the performance in 20th iteration from

Table 6: 20 iterations of rlCoP with policy and value guidance on the M2k dataset.

| Iteration | 1 | 2 | 3 | 4 | 5 | 6 | 7 | 8 | 9 | 10 |
|-----------|------|------|------|------|------|------|------|------|------|------|
| Proved | 1037 | 1110 | 1166 | 1179 | 1182 | 1198 | 1196 | 1193 | 1212 | 1210 |
| Iteration | 11 | 12 | 13 | 14 | 15 | 16 | 17 | 18 | 19 | 20 |
| Proved | 1206 | 1217 | 1204 | 1219 | 1223 | 1225 | 1224 | 1217 | 1226 | **1235** |

1235 to 1182. Replacing the final value with the best one from the value-only runs decreases the performance in 20th iteration from 1235 to 1144.

## 4.5 Evaluation on the Whole Miz40 Dataset

The Miz40 dataset is sufficiently large to allow an ultimate train/test evaluation in which rlCoP is trained in several iterations on 90% of the problems, and then compared to mlCoP on the 10% of previously unseen problems. This will provide the final comparison of human-designed proof search with proof search trained by reinforcement learning on many related problems. We therefore randomly split Miz40 into a training set of 29272 problems and a testing set of 3252 problems.

First, we again measure the performance of the unguided systems, i.e., comparing mlCoP, the bare prover and the non-learning rlCoP using only UCT with the simple goal-counting proof state evaluation heuristic. The results of runs with a limit of 200000 inferences are shown in Table 7. mlCoP here performs relatively slightly better than on M2k. It solves 13450 problems in total with a higher limit of $2 * 10^6$ inferences, and 13952 problems in total with a limit of $4 * 10^6$ inferences.

Table 7: Performance on the Miz40 dataset of mlCoP, the bare prover and non-learning rlCoP with UCT and simple goal-counting proof state evaluation. (200000 inference limit).

| System | mlCoP | bare prover | rlCoP without policy/value (UCT only) |
|--------|-------|-------------|---------------------------------------|
| Training problems proved | 10438 | 4184 | 7348 |
| Testing problems proved | 1143 | 431 | 804 |
| Total problems proved | 11581 | 4615 | 8152 |

Finally, we run 10 iterations of full rlCoP with UCT and both policy and value learning. Only the training set problems are however used for the policy and value learning. The inference limit is again 200000, the 0th run is as above, solving 7348 training and 804 testing problems. Table 8 shows the problems solved by iterations 1 to 10. rlCoP guided by the policy and value learned on the training data from iterations $0 - 4$ proves (in the 5th iteration) 1624 testing problems, which is 42.1% more than mlCoP run with the same inference limit. This is our final result, comparing the baseline prover with the trained prover on previously unseen data. 42.1% is an unusually high improvement which we achieved with practically no domain-specific engineering. Published improvements in the theorem proving field are typically between 3 and 10 %.

Table 8: 10 iterations of rlCoP with policy and value guidance on the Miz40 dataset. Only the training problems are used for the policy and value learning.

| Iteration | 1 | 2 | 3 | 4 | 5 | 6 | 7 | 8 | 9 | 10 |
|-----------|-------|-------|-------|-------|-------|-------|-------|-------|-------|-------|
| Training proved | 12325 | 13749 | 14155 | 14363 | 14403 | 14431 | 14342 | **14498** | 14481 | 14487 |
| Testing proved | 1354 | 1519 | 1566 | 1595 | **1624** | 1586 | 1582 | 1591 | 1577 | 1621 |

## 4.6 Comparison with State-of-the-art Saturation Systems

The Miz40 dataset was created [21] by running state-of-the-art first-order ATPs - mainly E [40] and Vampire [26] - on various axiom selections in many different ways, followed by *pseudominimization* [1] of the set of axioms. E and Vampire are complex human-programmed saturation-style ATP systems developed for decades. They consist of hundreds of thousands lines of code efficiently implementing many specific procedures and heuristics in low-level languages. This includes optimizations for equational reasoning based on term orderings, many heuristics for clause and literal selection, integration of propositional splitting, etc. The systems also include auto-configuration

methods, choosing portfolios of strategies. New strategies can be formulated in domain-specific languages either manually or, e.g., by evolutionary methods. The latter have been used in prior research to develop good strategies and their portfolios specifically targeting Mizar problems [46, 17].

Table 9 shows performance of E and Vampire run in several ways on the Miz40 dataset. In all cases we use a CPU limit of 3 seconds, which approximately corresponds to the time it takes to mlCoP to do 200000 inferences. Vampire is run in its default competition mode, where it uses a portfolio of strategies for each problem. E is run in four different ways: (*auto*) using its automated single-strategy selection, (*noauto*) using a default term ordering, only a simple clause selection, and no literal selection, (*restrict*) limiting additionally literal orderings, and (*noorder*), limiting all literal and term orderings. Since the majority of Miz40 problems contain equality, limiting the orderings often results in more prolific (less smart) generation of clauses by the inference rules.

Table 9: Performance of E and Vampire run in several ways on the Miz40 dataset.

| System | Vampire | E-auto | E-noauto | E-restrict | E-noorder |
|---|---|---|---|---|---|
| Training proved | 26304 | **26645** | 20914 | 11735 | 11235 |
| Testing proved | 2923 | **2942** | 2330 | 1271 | 1229 |
| Total proved | 29227 | **29587** | 23244 | 13006 | 12464 |

Both E and Vampire clearly outperform mlCoP and rlCoP when using their best strategies and portfolios. The performance drops when only weak clause selection and no literal selection is used. Further restricting literal and term orderings makes E weaker than rlCoP and comparable to mlCoP. This is consistent with the facts that (i) practically all Miz40 problems (32413) make use of equality, (ii) good strategies and term orderings have been invented for E and Vampire on the Mizar dataset for many years, and that (iii) the Miz40 dataset has been created as the Mizar problems that E or Vampire could solve.

The rlCoP implementation is two orders of magnitude smaller than E and Vampire: the core is about 2200 lines, and about 700 lines is the interface to the learners. An obvious question this performance comparison poses is how much can rlCoP improve solely by better machine learning in the current setting, and if further engineering and learning of more abstract guidance systems such as equality handling and literal/clause selection heuristics will be needed.

## 4.7 Examples

There are 577 test problems that rlCoP trained in 10 iterations on Miz40 can solve and standard mlCoP cannot.[7] We show three of the problems which cannot be solved by standard mlCoP even with a much higher inference limit (4 million). Theorem TOPREALC:10[8] states commutativity of scalar division with squaring for complex-valued functions. Theorem WAYBEL_0:28[9] states that a union of upper sets in a relation is an upper set. And theorem FUNCOP_1:34[10] states commutativity of two kinds of function composition. All these theorems have nontrivial human-written formal proof in Mizar, and they are also relatively hard to prove using state-of-the-art saturation-style ATPs. Figure 2 partially shows an example of the completed Monte-Carlo tree search for WAYBEL_0:28. The local goals corresponding to the nodes leading to the proof are printed to the right.

```
theorem :: TOPREALC:10
for c being complex number for f being complex-valued Function
 holds (f (/) c) ^2 = (f ^2) (/) (c ^2)

theorem :: WAYBEL_0:28
for L being RelStr for A being Subset-Family of L st
 ( for X being Subset of L st X in A holds X is upper )
 holds union A is upper Subset of L

theorem Th34: :: FUNCOP_1:34
```

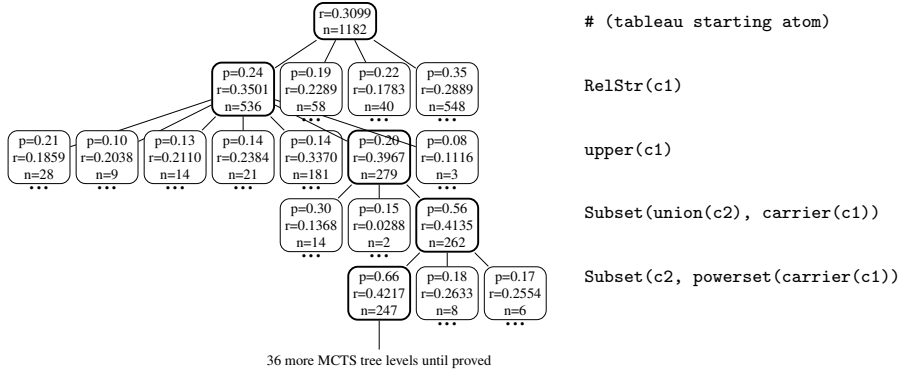

Figure 2: The MCTS tree for the `WAYBEL_0:28` problem at the moment when the proof is found. For each node we display the predicted probability $p$, the number of visits $n$ and the average reward $r = w/n$. For the (thicker) nodes leading to the proof the corresponding local proof goals are presented on the right.

```
for f, h, F being Function for x being set
 holds (F [;] (x,f)) * h = F [;] (x,(f * h))
```

# 5  Related Work

Several related systems have been mentioned in Section 1. Many iterations of a feedback loop between proving and learning have been explored since the MaLARea [47] system, significantly improving over human-designed heuristics when reasoning in large theories [22, 36]. Such systems however only learn high-level selection of relevant facts from a large knowledge base, and delegate the internal proof search to standard ATP systems treated there as black boxes. Related high-level feedback loops have been designed for invention of targeted strategies of ATP systems [46, 17].

Several systems have been produced recently that use supervised learning from large proof corpora for guiding the internal proof search of ATPs. This has been done in the connection tableau setting [48, 20, 10], saturation style setting [16, 30], and also as direct automation inside interactive theorem provers [12, 14, 49]. Reinforcement-style feedback loops however have not been explored yet in this setting. The closest recent work is [10], where Monte-Carlo tree search is added to connection tableau, however without reinforcement learning iterations, with complete backtracking, and without learned value. The improvement over the baseline measured in that work is much less significant than here. An obvious recent inspiration for this work are the latest reinforcement learning advances in playing Go and other board games [41, 43, 42, 3].

# 6  Conclusion

In this work we have developed a theorem proving algorithm that uses practically no domain engineering and instead relies on Monte-Carlo simulations guided by reinforcement learning from previous proof searches. We have shown that when trained on a large corpus of general mathematical problems, the resulting system is more than 40% stronger than the baseline system in terms of solving nontrivial new problems. We believe that this is a landmark in the field of automated reasoning, demonstrating that building general problem solvers for mathematics, verification and hard sciences by reinforcement learning is a very viable approach.

Obvious future research includes strong learning algorithms for characterizing mathematical data. We believe that development of suitable (deep) learning architectures that capture both syntactic and semantic features of the mathematical objects will be crucial for training strong assistants for mathematics and hard science by reinforcement learning.

## 7  Acknowledgments

Kaliszyk was supported by ERC grant no. 714034 *SMART*. Urban was supported by the *AI4REASON* ERC Consolidator grant number 649043, and by the Czech project AI&Reasoning CZ.02.1.01/0.0/0.0/15_003/0000466 and the European Regional Development Fund. Michalewski and Kaliszyk acknowledge support of the Academic Computer Center Cyfronet of the AGH University of Science and Technology in Kraków and their Prometheus supercomputer.

## Footnotes

[2]To minimize the required theorem proving background, we follow the more standard connection tableau calculus presentations using CNF and refutational setting as in [29]. The leanCoP calculus is typically presented in a dual (DNF) form, which is however isomorphic to the more standard one.

[3]We have used the RandomizedSearchCV method of Scikit-learn [35].

[4]We use L2-regularized L2-loss support vector regression and $\epsilon = 0.0001$ for LIBLINEAR.

[5]`https://github.com/JUrban/deepmath`

[6]Intel(R) Xeon(R) CPU E5-2698 v3 @ 2.30GHz with 256G RAM.

[7]Since theorem proving is almost never monotonically better, there are also 96 problems solved by mlCoP and not solved by rlCoP in this experiment. The final performance difference between rlCoP and mlCoP is thus $577 - 96 = 481$ problems.

[8]http://grid01.ciirc.cvut.cz/~mptp/7.13.01_4.181.1147/html/toprealc#T10

[9]http://grid01.ciirc.cvut.cz/~mptp/7.13.01_4.181.1147/html/waybel_0#T28

[10]http://grid01.ciirc.cvut.cz/~mptp/7.13.01_4.181.1147/html/funcop_1#T34

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
