[Reviews · NeurIPS 2018]

Reviewer 1



This paper presents a theorem proving algorithm that leverages Monte-Carlo simulations guided by reinforcement learning from previous proof searches. In particular, to guid the search, the authors use the UCT formula augmented by (1) learning prior probabilities of actions for certain proof states (policy learning), and (2) learning the values corresponding to the same proof states (policy evaluation). The proposed algorithm is using no domain engineering. The authors evaluate their proposal on two datasets: Miz40 and M2k. The evaluation shows the proposed solution outperforms state-of-the are by solving ~40% more problems. This is a solid paper: it addresses a hard problem, and provide solutions that show significant improvements over the prior work. I have a few questions however, which mainly seek clarifications of the evaluation results: - You mention "We have also experimented with PUCT as in AlphaZero, however the results are practically the same." Can you provide some more color on this? How close were the results? Any tradeoffs? - Does rlCoP with policy and value guidance solve all the problems mlCoP solves on Miz40 dataset? If not, which problems does mlCoP solve and rlCoP doesn't? Same questions for rlCop with policy guidance vs mlCoP for the M2k dataset. - Is there anything special about the solutions shown in Sec 4.6 besides the fact they are more involving? It would be nice to provide some more intuition of why your algorithm perform well in those cases. - When you are training rlCoP on the Miz40 dataset how do you pick the problems in the training set? Randomly? How robust are the results to picking a different training set? Minor: - "Published improvements in the theorem proving field are typically between 3 and 10 %." Can you please give some references to support your claim. - Please user either Mizar40 or Miz40 for consistency.

Reviewer 2



The paper uses reinforcement learning for automated theorem proving. This is a timely and important topic. The NIPS community probably does not know much about theorem proving and hence the paper may be hard to follow for the average NIPS participant. However, I've long waited for someone to try and use the recent progress we have seen in RL for ATP (which I find much more interesting and important as an application compared to games) and hence I think this paper will make a nice contribution for NIPS. Having said that I'm still a little puzzled by the description of the features and the state space. I would have liked more details here and an explanation for how these features are used in standard ATP systems. If you're bootstrapping on features that have been proven useful in non-ML/non-RL ATP systems with hand-crafted selection heuristics, this should be explained very clearly. Also, it would have been nice to at least mention in passing results for other standard theorem provers for this standard data set (e.g. E or Vampire) for proper calibration as to whether this relative progress that you report is likely to translate into absolute progress for the field of ATP. Update: The authors have commented on my two main concerns and if they add the relevant discussions/results to the paper it will be a stronger paper.

Reviewer 3



This paper develops a new theorem proving algorithm that uses Monte-Carlo search guided by reinforcement learning from previous proof searches. The algorithm shows strong performance as compared with a connection tableau system. The paper is well-written and interesting. The proposed approach requires little domain engineering and seems to be effective, but I'm not familiar with existing works in ATP, and will find a discussion on the state-of-the-art solvers and existing performance on Mizar40 to be helpful. Overall, the approach seems promising and may have the potential to be applied to solve general mathematical problems. *** After rebuttal I've read other reviews and the author's response and have not changed my opinion.